# Metabolomics Benefits from Orbitrap GC–MS—Comparison of Low- and High-Resolution GC–MS

**DOI:** 10.3390/metabo10040143

**Published:** 2020-04-04

**Authors:** Daniel Stettin, Remington X. Poulin, Georg Pohnert

**Affiliations:** Institute for Inorganic and Analytical Chemistry, Bioorganic Analytics, Friedrich Schiller University Jena, 07743 Jena, Germany; daniel.stettin@uni-jena.de (D.S.); remington.poulin@uni-jena.de (R.X.P.)

**Keywords:** Orbitrap Gas Chromatography–Mass Spectrometry (Orbitrap GC–MS), high-resolution mass spectrometry (HRMS), metabolite identification, instrument comparison, comparative metabolomics, diatom, osmotic stress

## Abstract

The development of improved mass spectrometers and supporting computational tools is expected to enable the rapid annotation of whole metabolomes. Essential for the progress is the identification of strengths and weaknesses of novel instrumentation in direct comparison to previous instruments. Orbitrap liquid chromatography (LC)–mass spectrometry (MS) technology is now widely in use, while Orbitrap gas chromatography (GC)–MS introduced in 2015 has remained fairly unexplored in its potential for metabolomics research. This study aims to evaluate the additional knowledge gained in a metabolomics experiment when using the high-resolution Orbitrap GC–MS in comparison to a commonly used unit-mass resolution single-quadrupole GC–MS. Samples from an osmotic stress treatment of a non-model organism, the microalga *Skeletonema costatum*, were investigated using comparative metabolomics with low- and high-resolution methods. Resulting datasets were compared on a statistical level and on the level of individual compound annotation. Both MS approaches resulted in successful classification of stressed vs. non-stressed microalgae but did so using different sets of significantly dysregulated metabolites. High-resolution data only slightly improved conventional library matching but enabled the correct annotation of an unknown. While computational support that utilizes high-resolution GC–MS data is still underdeveloped, clear benefits in terms of sensitivity, metabolic coverage, and support in structure elucidation of the Orbitrap GC–MS technology for metabolomics studies are shown here.

## 1. Introduction

Metabolite annotation is an important step in any untargeted mass spectrometry (MS)-based metabolomics study, allowing for chemically and biologically sound interpretation of analytical data [1]. In current studies, usually less than 30% of compounds are identified because the coverage of existing molecules in mass spectral libraries is limited [2]. The identification of an unknown, either by MS experiments or alternative techniques such as nuclear magnetic resonance spectroscopy (NMR), has been the most time-consuming step of metabolomics research for more than a decade, requiring labor-intensive manual interpretation of spectral data by an expert [3]. However, the field is currently entering a new era of metabolite annotation, where novel analytical technologies, computational algorithms and community driven open database resources will enable the rapid annotation of whole metabolomes [4].

The Orbitrap mass spectrometer coupled with liquid chromatography (LC) has contributed to this development by providing high mass resolving power combined with tandem MS capabilities [5]. MS^2^ spectra recorded with high-resolution improve database annotations providing that spectra in the databases are deposited in high resolution as well. This is the case for the Human Metabolome Database (HMDB) [6], MassBank [7] and Global Natural Products Social Molecular Networking (GNPS) [8]. High-resolution data also improve annotation attempts even when no direct spectral match is available. Many computational tools enabling this have emerged in recent years, such as MetFrag [9], SIRIUS and CSI:FingerID [10], MS-FINDER [11] or CFM-ID [12].

In contrast, the Orbitrap mass spectrometer coupled with gas chromatography (GC) [13] has seen only little use in metabolomics research since its introduction [14,15,16,17,18,19]. This might be because LC separation is described to provide the best compromise between biggest metabolome coverage and simplest sample treatment and thus serves as the go-to technique for MS-based metabolomics [20,21]. However, GC–MS has several advantages over LC–MS [22]. For example, ionization efficiencies in Electrospray ionization (ESI) are compound dependent and can vary in the range of 0–100% [23]. MS^2^ spectra can vary greatly depending on the type of mass spectrometer used [24]. Compared to LC–MS, GC–MS is a much more robust technique, providing reproducible retention times, universally applicable retention indices, and better resolved peaks. Spectra show highly reproducible fragmentation and, consequently, more powerful database support [25], with the biggest databases being the National Institute of Standards and Technology (NIST) Mass Spectral Library, with over 250,000 spectra, and the Wiley Registry, with over 700,000 spectra.

Another possible reason for the lack of interest in the Orbitrap GC–MS in metabolomics is that making use of high-resolution GC–MS data is not straight forward. Almost all available GC–MS spectral libraries carry only unit-mass data, making high-resolution data unnecessary for their use. Further, it is difficult to apply annotation tools developed for high-resolution LC–MS to high-resolution GC–MS data. In principle, all of them follow the same workflow: the measured accurate mass of an unknown is queried against structure databases, then in silico predicted properties (fragmentation pattern, substructures, etc.) are compared between obtained candidate structures and the unknown. A scoring system helps in evaluating the likelihood of a putative annotation. This infers several obstacles when applied to GC–MS data. Electron ionization (EI) spectra often lack a molecular ion [26]. Due to the unique mode of measurement in Orbitrap MS (storage of ions in a linear ion trap prior to injection into the actual Orbitrap), ions at very low abundance, like most molecular ions, are lost, further increasing this problem. In addition, EI fragment spectra differ from ESI–MS^2^ spectra and thus require adjustments to in silico interpretation or prediction [27]. Collision-induced dissociation (CID) of the even electron ions produced by ESI yields mostly heterolytic fragmentation remote to the charge site or with the charge migrating to a new site, resulting in new even electron fragments [24]. The much more unstable (odd electron) radical cation formed by EI leads to a variety of homo- and heterolytic cleavages and rearrangements, usually originating from the site of the charge or radical, resulting in both even and odd electron fragments [26], complicating simulation.

However, strategies addressing these problems do exist. High-resolution data has been used before to enhance unit-mass database matching with a method named “high-resolution filtering” [28]. This operates by determining the sum formulas of all fragments found in the EI spectrum and determining whether they form a subset of the sum formula of a database hit. Thereby, plausibility of a database hit can be tested efficiently. Chemical ionization (CI) can be used to generate molecular ions for sum formula calculation [29]. This results in two separate datasets, one with possible molecular ions (CI) and one with spectra rich in fragments, potentially lacking the molecular ion (EI). Sum formulas from CI data can be used to generate candidate structures with the help of structural databases. Complexity is added by the fact that GC metabolomics samples are derivatized with methoxylating and silylating agents prior to analysis [30], and so candidate structures must be in silico derivatized before comparison.

Computational tools exist that can partly circumvent these problems and make use of high-resolution GC–MS data. They have been successfully applied for identification of unknowns in a small number of studies. Lai et al. used a combination of MS-FINDER [11] and CFM-ID [31] to identify eight unknowns from previous studies of cancer cells, *Escherichia coli*, *Chlamydomonas* and *Artemisia* [32], as well as five unknowns found in a great number of studies including human, animal and marine samples [33]. Qui et al. used CFM-ID to identify one previous unknown from *Saccharomyces cerevisiae* [17].

Despite the availability of these methods, utilization of high-resolution GC–MS technology in metabolomics research is still rare. In theory, the identification of unknowns is supported but the absolute gain of information when employing high-resolution GC–MS in comparison to established unit-mass analyzers is not fully explored. In general, studies gauging the benefit of high- versus low-resolution systems are scarce in the field. Some ring trials have been conducted, comparing different laboratories with different mass spectrometers, chromatographic systems, data analysis pipelines, etc., with the aim to validate the hypothesis-generating power of metabolomics as a whole [34,35]. However, those have too many interfering factors to be able to attribute differences found to the resolution of the mass spectrometer used.

In other fields, such as residue analysis, detailed comparisons of mass spectrometers are more common [36] and the replacement of established platforms with novel technology is carefully evaluated [37]. Interestingly, such comparisons have uncovered correlations between instrument resolution and false results depending on the complexity of the sample matrix [38,39]. Consequently, studies like this also serve as a guideline for which instrument would be the best suited for different analytical tasks in respect to the monetary investment required.

Therefore, the present study aimed to compare total results of a metabolomics experiment put through a high- as well as a low- mass resolution analysis workflow. Orbitrap GC–MS was chosen to represent the high-resolution platform. The single-quadrupole ISQ GC–MS was chosen to represent the low-resolution platform since this instrument is frequently used in recent metabolomics research [40,41,42]. We investigated the response of a non-model organism, the diatom *Skeletonema costatum*, to osmotic stress compared to a control. Differences in metabolic coverage, results of statistical analysis and compound annotation were analyzed for both workflows.

## 2. Results

### 2.1. Increased Metabolic Coverage Using a Smaller Sample

Preliminary experiments revealed different sensitivities of the two platforms used (Figure 1). When using the same sample concentration with minimal split ratios on both systems, peak picking with XCMS [43] with a signal-to-noise threshold of 3 resulted in 8850 and 41,588 *m*/*z* features for the ISQ GC–MS and Orbitrap GC–MS, respectively. To maximize data obtained from both systems, sample concentrations were subsequently increased for ISQ measurements (cultivation flasks had to be changed from 40 mL per sample to 400 mL per sample to obtain enough biological material). Split ratios were adjusted until the highest peak in a pooled sample was approximately 1E9 total ion counts (TIC), which is a good value for gaining maximum signals without detector or column overload or damage to the filament. For overall similar TIC on both instruments, an 8-fold more concentrated sample had to be used on the ISQ GC–MS (Figure 2). Peak picking with a signal-to-noise threshold of 3 resulted in 23,366 and 45,714 *m*/*z* features in the ISQ GC–MS and Orbitrap GC–MS, respectively. After grouping fragment features into compounds and filtering irrelevant data (missing values, compounds present in media blanks), 114 (ISQ GC–MS) and 339 compounds (Orbitrap GC–MS) remained—this was a 3-fold difference (Appendix A).

### 2.2. Orbitrap and Single-Quadrupole Systems Detect Different Biomarkers

Significantly dysregulated compounds between controls and salinity stressed algae in both datasets were compared to assess differences in biomarker findings between both methods. Interestingly, the significant compounds of both datasets are only 28% overlapped, with 9 compounds found in the ISQ GC–MS dataset that are not considered significant in the Orbitrap GC–MS dataset and 46 compounds vice versa (Figure 3). For the compounds missing from the ISQ GC–MS results, it was found that in four cases, the compound was missing due to faulty deconvolution. In 12 cases, *p*-value or fold change did not meet the threshold criteria (*p*-value < 0.05, fold change > 2) for significant compounds. In 29 cases, there were too few or no fragments detected by the ISQ GC–MS. The Orbitrap GC–MS thus detects more compounds because of its lower detection limit. Further, two of the nine compounds missing from Orbitrap GC–MS results were excluded because they were present in media blanks. In two cases, the compound is missing due to faulty deconvolution. The remaining five compounds were missing because *p*-value or fold change did not meet the threshold criteria for significant compounds. Manual reintegration of peak areas revealed this as neither a result of multiple testing correction nor processing artifacts.

### 2.3. Similar Statistical Discrimination of Samples on Both Platforms

Multivariate statistics were employed to determine whether both high- and low-resolution methods would be able to assign samples into the correct experimental groups. This is important both for the purpose of diagnostic metabolomics (classifying unknown samples) and as verification that the experiment induced a metabolic response in the investigated organism.

Principal component analysis score plots (PCA) (Figure 4) show a similar discriminatory power of both methods. The first two principal components add up to 76.1% in the ISQ GC–MS dataset and 75.1% in the Orbitrap GC–MS dataset, respectively. Even though the number of compounds included in the statistical analysis in each dataset varies greatly (104 in the ISQ GC–MS dataset vs. 322 in the Orbitrap GC–MS dataset), the difference in explained variability by the first two principal components is negligible. Since data has been autoscaled, high and low concentrated compounds contribute equally to the displayed separation.

### 2.4. High-Resolution Data Supports Spectral Database Matching

Matching EI fragment spectra to libraries is a fast way to (in case of high matching scores) confidently annotate metabolites. Because available libraries only carry unit-mass resolution data, high-resolution data per se does not enhance the matching process. However, sum formulas of fragment ions can be calculated from raw data with high-resolution MS, supporting or falsifying a library match.

Mass spectra from all compounds were compared to EI database spectra and hits were manually curated for plausibility using criteria of matching score, peak number, and matching patterns. Out of the 339 compounds initially detected by the Orbitrap GC–MS, 182 were only present in trace amounts with less than 20 fragments. Of the remaining 157 compounds, a substantial proportion displayed insufficient matching scores for an assignment (Figure 5). In such cases, verification of library matches with high-resolution data solidified the inappropriate assignment of the database, confirming the nature of the given compound as a true unknown. Additionally, it was found that in two cases out of 75 library annotations from the Orbitrap GC–MS dataset, top hits were disproven with accurate mass information. In general, the complexity of unit-mass fragmentation spectra turned out to be sufficient to annotate compounds correctly, if they are present in the spectral database, but the aspect of additional verification using HRMS is lacking. Thus, high-resolution filtering enhances the reliability of library annotations by providing an additional criterion to support plausibility of respective hits.

The question arose whether overall matching scores with Orbitrap GC–MS data decrease because of the Orbitrap’s unique mode of measurement that causes deviations of the relative peak intensities compared to conventional spectra. However, an overall comparison of matching scores between the two systems was not feasible because frequent deconvolution artifacts (single fragments missing from a deconvoluted spectrum or additional fragments presents from a co-eluting compound) have a bigger influence on matching scores than the deviations introduced by the Orbitrap. Considering individual high-quality spectra taken directly from raw data, the Orbitrap GC–MS indeed provides slightly lower matching scores (Appendix A).

### 2.5. High-Resolution Data Enabled Identification of One Unknown

The potential to rapidly identify unknowns is one of the main motivations behind investing high-resolution MS equipment. Possible molecular formulas were generated for significantly dysregulated unknowns using chemical ionization and high-resolution MS. Candidate structures were then retrieved from all databases embedded in the tool MS-FINDER. In silico fragmentation of candidates was carried out both with MS-FINDER and CFM-ID. Results were compared with the recorded EI–MS spectrum using the high-resolution data of the fragments. One unknown was identified this way as dehydroascorbate, and its identity was confirmed with an analytical standard.

## 3. Discussion

Few studies in the field of metabolomics have aimed to directly compare results obtained from high- and low-resolution instrumentation [19,44,45]. They describe a general increase in features/compounds detected due to the enhanced selectivity and sensitivity of high-resolution MS. To make full use of the range of the respective detectors, the low-resolution ISQ GC–MS required an 8-fold higher sample concentration. Available amount of sample is usually not a problem for environmental samples or organisms with scalable cultivation but might be a limiting factor in human or animal studies. Even more so, at an 8-fold lower sample concentration, the Orbitrap GC–MS still yielded almost 3-fold more detected metabolites.

Interestingly, both low- and high-resolution pathways resulted in similar multivariate predictive power to separate samples into the respective experimental groups. This is on par with previous publications conducting metabolomics experiments on different mass spectrometers [45,46,47]. However, it has to be noted that the experiment employed here induces a strong metabolic change in the investigated organism. In a setting where this is not a given, access to lower concentrated compounds that are covered by the Orbitrap GC–MS platform may be needed for sufficient separating power.

Another similarity with previous studies is the fact that while multivariate predictive power is equal, only a limited overlap of biomarkers between both systems was found. Gika et al. connected a single LC system to a quadrupole time-of-flight (QTOF) MS and an ion-trap MS simultaneously but still only found a 23% overlap of dysregulated ions [45], which is in line with our reported 28% overlap. Glauser et al. found an 81% overlap between a high-resolution LC-QTOF and Orbitrap LC [46] and Cajka et al. found an overlap of 92% on nine LC–MS systems but only considering significant compounds that were already common to at least two systems [47]. These studies suggest that independent from chromatographic performance, the MS used has a profound impact on the results. This raises general concerns about the validity of specific biomarker findings in metabolomics studies conducted on a single system. Significant effort has been dedicated to statistically trying to eliminate false positives and negatives from results [48] but bias introduced by MS instrumentation is generally not investigated in metabolomics studies.

It has to be mentioned that the ISQ GC–MS used in this study was purchased in 2011 and has been in frequent use since, resulting in a diminished performance. In repeated measurements of a fatty acid methyl ester (FAME) mixture, it became apparent that the relative standard deviations (RSD) in the ISQ GC–MS gradually worsened for FAMEs with higher retention times (Appendix A). Increasing the injector temperature alleviated this problem in FAME measurements, resulting in the explanation that at 250 °C, the injector in the ISQ GC–MS has problems evaporating higher boiling compounds consistently. This problem is not apparent in the measured metabolomics dataset though (Appendix A). It is for these reasons that results presented here connected to the sensitivity and reproducibility (*p*-values for significant hits) of biomarker findings cannot be generalized to new, state-of-the-art single-quadrupole GC systems. In fact, modern quadrupole mass analyzers may provide advantages over the Orbitrap GC–MS such as an increased scan rate. This is not assessed nor discussed here. However, as stated above, single-quadrupole GC–MS systems such as the one used in this study are still being used in many labs for relevant metabolomics studies [40,41,42], which justifies this comparison. Data evaluation relying only on the ISQ GC–MS data misses multiple annotated metabolites from the significantly dysregulated compounds, either because of too high in-group variability, too low concentration or the inability to annotate the compound without high-resolution data. It is, therefore, a multitude of factors that make the Orbitrap GC–MS more performant for untargeted metabolomics experiments in this comparison.

The result that can be generalized from this study is how much high-resolution data improves annotation attempts. Sum formula calculation of fragments provides an additional criterion verifying or falsifying hits with conventional EI spectral library matching, which turned out to be relevant in a few cases in this study. Additional methods such as support of EI data with CI mass spectra, specialized databases and in silico fragmentation algorithms improve annotations of unknowns as demonstrated here. Employing such a workflow and manual data curation, the Orbitrap GC–MS enabled the identification of one significantly dysregulated unknown, a feature that would have been impossible to explore using only low-resolution MS data. Despite this seemingly low benefit of high-resolution GC–MS in terms of identification of unknowns, there is substantial potential once advanced computational tools supporting structural elucidation or dereplication are available. At the moment, predicted sum formulas of fragments by high-resolution data are, on their own, insufficient to obtain meaningful information about unknowns. Suitable software developments will surely support the dereplication of unknowns based on high-resolution GC–MS fragment data, as has been achieved for LC–MS with the molecular networking approach [8]. Computational tools dealing with LC–MS data have been increasing in number in recent years and research groups are now in constant competition to improve the quality of annotation [49]. Considering more than half of the non-trace compounds detected by the Orbitrap GC–MS remained unidentified, this task becomes more important, with each generation of MS technology pushing towards lower detection limits.

This study shows that, in respect to the monetary investment required to upgrade from existing instrumentation to high-resolution instrumentation, the “better” choice is dependent on the analytical task at hand. The low-resolution ISQ GC–MS turned out to be sufficient when metabolic changes need to be analyzed only on a multivariate level, or the identification of only higher abundant primary metabolites (present in EI libraries) is of concern. If one aims to maximize the information obtained from metabolomics samples in terms of metabolic coverage and annotation of unknowns, high-resolution data enabled by the Orbitrap GC–MS should be the preferred choice.

## 4. Materials and Methods

### 4.1. Instrumentation

Instruments used were the Q-Exactive™ Orbitrap GC ™ system, including a Q-Exactive™ Orbitrap™ mass spectrometer connected to a Trace™ 1310 GC equipped with a TriPlus™ RSH™ Autosampler (Thermo Scientific, Bremen, Germany) and the GC ISQ™ system, including a ISQ™ single-quadrupole mass spectrometer connected to a Trace™ GC Ultra equipped with an AS 3000 autosampler (Thermo Scientific, Bremen, Germany).

### 4.2. Standards

The following chemicals were used as analytical standards: citric acid, dehydroascorbic acid, eicosapentaenoic acid, d-pyroglutamic acid, l-threono-1,4-lactone (Sigma-Aldrich, Munich, Germany), l-arginine monohydrochloride, l-glutamic acid, l-isoleucine, l-leucine, l-lysine, l-phenylalanine, l-proline, l-tryptophane, l-tyrosine, l-threonine, l-valine (Fluka Analytical, Munich, Germany), d-(+)-glucose (VWR chemicals, Darmstadt, Germany), l-ornithine monohydrochloride (abcr), and pyrrole-2-carboxylic acid (Acros Organics, Darmstadt, Germany).

### 4.3. Cultivation

*S. costatum* (RCC75) was obtained from the Roscoff Culture Collection (Roscoff, France). Artificial sea water medium (400 mL each) [50] was inoculated with 8 × 10^6^ cells per culture and grown at 13 °C under a 14/10 h light/dark regime with an illumination of 15–24 µmol photons m^−2^ s^−1^ for 15 days as standing cultures. At day 15, either 50 mL of culture medium (controls) or culture media enriched with 0.225 g/mL sodium chloride (Carl Roth, Karlsruhe, Germany) for the osmotic stress treatment was added. This treatment resulted in a salinity of 35 practical salinity units (PSU) in the control group and 60 PSU in the osmotic stress treatment group, respectively. After 27 h, cell densities of all batch cultures were measured by counting under a LEICA DM 2000 light microscope (LEICA, Wetzlar, Germany) in a Fuchs-Rosenthal counting chamber and cells were harvested for extraction. Culture media blanks for both control and treatment conditions were also generated.

### 4.4. Extraction

All cells were concentrated on GF/C filters with a pore size of 1.2 µm (Whatman, Dassel, Germany) under a reduced pressure of 700 mbar, and filters were immediately transferred into the extraction mix consisting of chilled methanol (LiChrosolv, Merck, Darmstadt, Germany), ethanol (LiChrosolv, Merck, Darmstadt, Germany) and chloroform (HPLC gradient grade, Fisher Chemical, Thermo Scientific, Bremen, Germany) 1:3:1 (*v*:*v*:*v*) [51]. After ultrasonication for 5 min, extracts were centrifuged at 30,000× *g* for 15 min. The supernatants were stored at −80 °C.

### 4.5. Sample Workup

Volumes equivalent to 5 × 10^6^ cells (Orbitrap GC) or 4.2 × 10^7^ cells (ISQ GC–MS) per sample were taken from each extract. Both cell numbers are below the recommended maximum cell number in our validated in-lab standard operating procedure (SOP) that has been published [51]. Pooled quality control (QC) samples were prepared by combining equal volumes from each extract [52]. All samples were dried under vacuum and reconstituted in 50 µL pyridine (CHROMASOLV Plus, Sigma-Aldrich, Munich, Germany) containing 20 mg/mL methoxyamine monohydrochloride (Sigma-Aldrich, Munich, Germany). Samples were heated to 60 °C for 1 h and stored at room temperature overnight. A volume of 50 µL of N,O-bis(trimethylsilyl)trifluoroacetamide (BSTFA) (Thermo Scientific, Bremen, Germany) was added to each sample, and all samples were heated to 60 °C for 1 h. A C_7_–C_40_ alkane standard mix (Supelco, Munich, Germany) was added to one QC sample.

### 4.6. Data Collection

Both GC–MS used were equipped with (lightly used) Zebron ZB-SemiVolatiles columns (30 m × 0.25 mm × 0.25 µm, Phenomenex, Aschaffenburg, Germany). Regenerated (CS-Chromatographie Service, Langerwehe, Germany) thermo liners were used for the experiment (78.5 mm in length, 4 mm in inner diameter, straight split with quartz wool in Orbitrap GC–MS; 105 mm in length, 5 in mm inner diameter, straight split with quartz wool in GC-ISQ). New septa (Thermo, Bremen, Germany) were used. The same oven program was used: the initial temperature 80 °C was maintained for 2 min, raised to 120 °C at a rate of 20 °C/min, maintained for 1 min, raised to 250 °C at a rate of 5 °C/min, raised to 320 °C at a rate of 10 °C/min and maintained for 2 min. The ISQ GC–MS was run with a carrier gas flow of 1 mL/min, a split ratio of 1:50, an injector temperature of 250 °C and a solvent delay of 5.7 min. The injection volume was 1 µL. The transfer line was kept at 250 °C and the ion source at 280 °C. Data was recorded in full scan centroid mode at 50–600 *m*/*z* with a scan rate of 2 Hz. The Orbitrap GC in the EI mode was run with a carrier gas flow of 1 mL/min, a split ratio of 1:50, an injector temperature of 250 °C and a solvent delay of 5.7 min. The injection volume was 1 µL. The MS transfer line was kept at 250 °C, the auxiliary heaters at 280 °C, and the ion source at 300 °C. The automatic gain control was set to 10^6^ and the maximum injection time to auto. Data was recorded in full scan profile mode at 50–600 *m*/*z* with a Fourier transform resolution of 120,000, resulting in a scan rate of approximately 3.8 Hz. In the CI mode, the split ratio was changed to 1:5, the injection volume to 2.5 µL and the ion source temperature to 180 °C. The full scan range was set to 80–1000 *m*/*z*. Methane (N55, Air Liquide, Düsseldorf, Germany) with a flow rate of 1.5 mL/min was used as the ionization gas. Study files can be found at the Metabolights repository www.ebi.ac.uk/metabolights/MTBLS1104.

### 4.7. Data Preprocessing

Raw files were converted to mzXML format using the tool MSConvert from the Proteowizard Suite (proteowizard.sourceforge.net). In case of profile Orbitrap GC data, the peak picking routine provided by the vendor was enabled. Data were preprocessed using a custom R pipeline (Appendix A). The package XCMS [43] was used for *m*/*z* feature deconvolution and integration, the package CAMERA [53] was used for grouping features into compounds and a custom script together with the package metaMS [54] was used for extracting compound pseudospectra as NIST compatible .msp files. Signal drift correction was carried out with the package statTarget [55]. Compounds exceeding an RSD of 20% after signal drift correction in QC samples were excluded. Compounds present in culture medium blanks not displaying an at least a 5-fold higher average peak area in at least one experimental group (osmotic stress or control) were excluded.

### 4.8. Statistical Analysis

Statistical analysis was carried out via Metaboanalyst 4.0 web [56,57]. Data was log transformed to correct for heteroscedasticity and autoscaled so that only correlations of relative changes in concentration are analyzed [48]. Univariate *p*-values were adjusted with Metaboanalyst’s false discovery rate correction.

### 4.9. Identification Workflow

The identification workflow for every compound consisted of the following steps (Figure 6). Step 1: Extracted pseudospectra were compared to database spectra with dot-product scoring using the NIST Mass Spectral Search Program v.2.0g (National Institute of Standards and Technology, USA), Identity, Normal search with no constraints. Databases used were the NIST/EPA/NIH/ EI Mass Spectral Library NIST 11 (copyright United States Department of Commerce)mainlib and replib, Golm metabolome database [58] version 21.11.2011 (gmd.mpimp-golm.mpg.de) and Fiehnlib [59] obtained from MassBank of North America (mona.fiehnlab.ucdavis.edu). All matches were manually checked for plausibility because extracted spectra with a low number of signals can generate high scores even when just very few peaks match to a database spectrum. Step 2: Since the automated application of “high-resolution filtering” [28] is not openly available, database matches were manually evaluated. Sum formulas of the most intense and highest *m*/*z* ions of compounds putatively annotated in the previous step were calculated using the Orbitrap GC’s high-resolution data and compared to the sum formula of the proposed compound. To exclude false comparisons due to deconvolution artifacts, chromatographic peak shapes of all relevant ions were compared to compound peak shapes in the raw data. An annotation was deemed as confirmed by high-resolution data when each sum formula obtained this way represented the whole or a subset of the proposed compound’s sum formula. Step 3: Molecular ions of remaining unknowns were obtained by CI measurements. Molecular ions were assigned to corresponding EI spectra by searching the CI chromatogram for common fragments and similar sum formulas at the retention index of the compound detected in the EI mode. Identity as a molecular ion was confirmed by the presence of the ion series [M−CH_3_]^+^, [M+H]^+^, [M+C_2_H_5_]^+^ and [M+C_3_H_5_]^+^ [30]. High-resolution centroided pseudospectra and molecular ion masses were fed into MS-FINDER version 3.12. Settings used were the following. Mass spectrum: Mass tolerance of 3 ppm in MS1 and MS2 and a relative abundance cutoff of 5%. Formula finder: LEWIS and SENIOR checked, element ratio check at 99.7%, element probability checked. Element selection and number of TMS and MeOX groups were changed every query depending on preliminary information gained from the high-resolution EI spectrum and isotope pattern. Structure finder: Tree depth 3, fragmentation library for EI checked, and maximum report number 100. Data source: all local databases checked, MINE and PubChem always used. Highest scoring structures for each query were subjected to in silico fragmentation via the CFM-ID 2.0 webpage (cfmid.wishartlab.com). A structure was considered as a valid annotation if at least one fragmentation pattern (received by either MS-FINDER of CFM-ID) showed similarities to the recorded EI spectrum and a spectral record of the compound did not already exist in the EI databases. All annotations obtained by this workflow were validated by measuring authentic analytical standards. Additional peaks appearing in the analysis of pure standards (e.g., partly derivatized versions) were also compared with remaining unknowns. Compound identity was confirmed by manual comparison of EI accurate mass spectra and matching Kovats indices.

## Figures and Tables

**Figure 1 metabolites-10-00143-f001:**
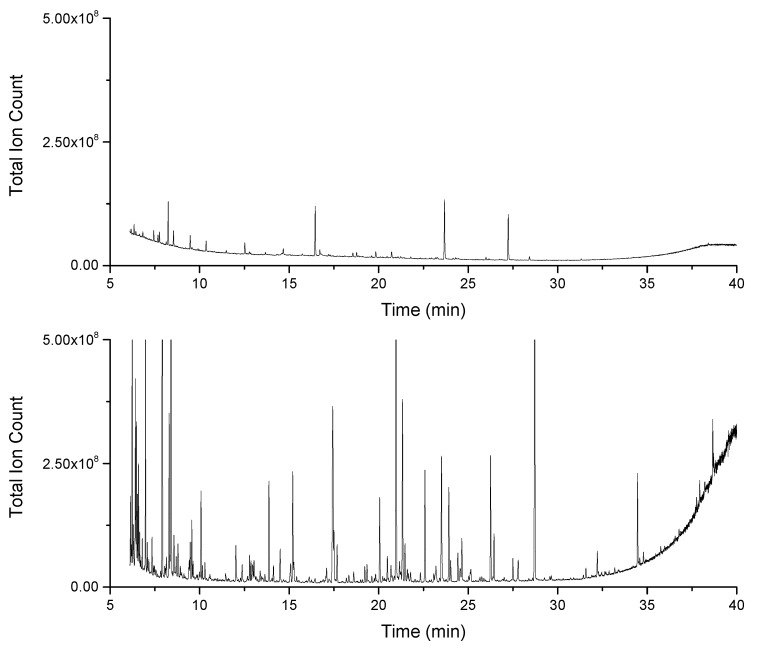
Chromatograms from preliminary experiments before sample concentration adjustment. Chromatograms show derivatized *Skeletonema costatum* extracts with a concentration equivalent to 2.5 × 10^3^ cells per µL sample measured on the Orbitrap GC (bottom), with a split ratio of 1:5, and the GC-ISQ (top), with a split ratio of 1:8 (minimal split ratios on each machine).

**Figure 2 metabolites-10-00143-f002:**
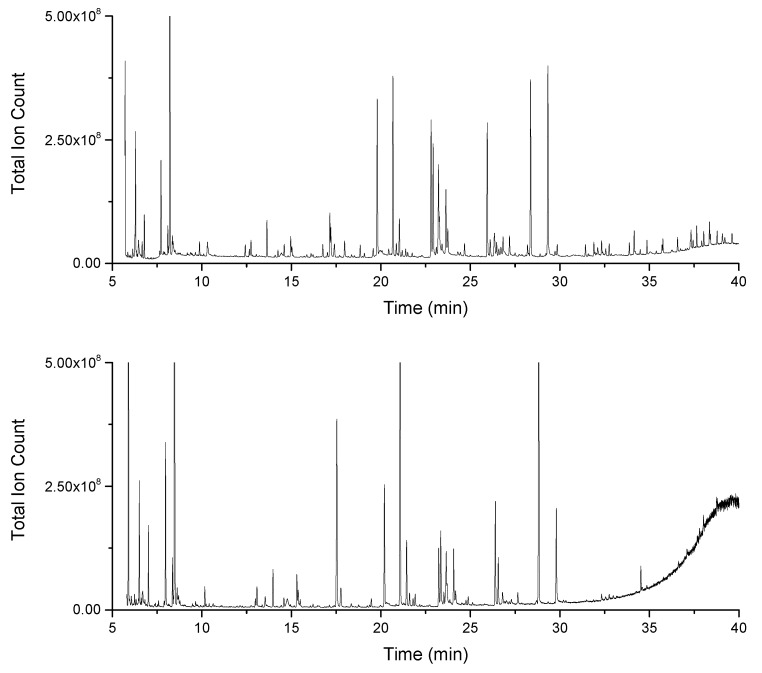
Chromatograms of pooled samples of the salt stress experiment. Chromatograms show *SSkeletonema costatum* extracts measured on the ISQ GC–MS (top) and on the Orbitrap GC–MS (bottom). Orbitrap data were recorded with a split ratio of 1:50 and a concentration equivalent to 5 × 10^4^ cells per µL of sample, GC-ISQ data with a split ratio of 1:50 and a concentration equivalent to 4 × 10^5^ cells per µL of sample.

**Figure 3 metabolites-10-00143-f003:**
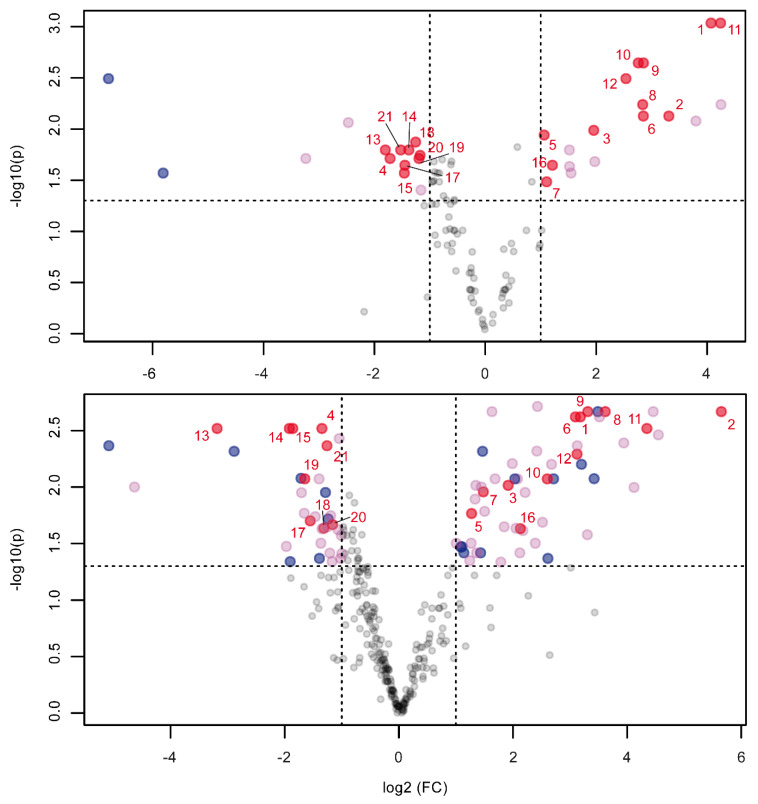
Volcano plot analysis of the comparative metabolomics experiment on salinity stress of microalgae. Samples were split and measured on both the unit-mass ISQ GC–MS (top) and the high-resolution Orbitrap GC–MS (bottom). Compounds exceeding a fold change (FC) of 2 (peak area from control to stressed) and below a *p*-value of 0.05 are deemed significantly dysregulated and are plotted in red. Marked in blue are processing artifacts that were not discernible from the background noise upon manual inspection. This left 30 compounds in the ISQ GC–MS and 68 compounds in the Orbitrap GC–MS dataset, respectively. Highlighted in red and numbered are the same compounds found to be significant in both datasets.

**Figure 4 metabolites-10-00143-f004:**
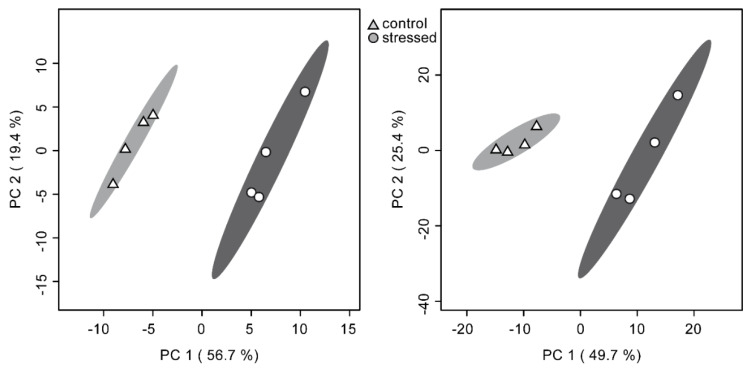
Principal component analysis (PCA) of the comparative metabolomics experiment with samples split and measured on both the unit-mass ISQ GC–MS (left) and the high-resolution Orbitrap GC–MS (right). Data shown was log transformed and autoscaled. There is no discernible difference between both low- and high-resolution systems concerning the ability to separate samples into experimental groups in this study.

**Figure 5 metabolites-10-00143-f005:**
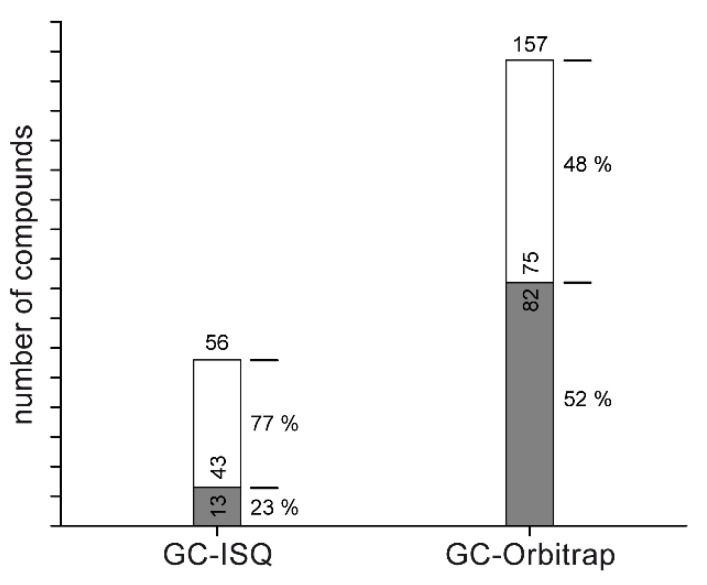
Ratio of compound annotation in each dataset. Samples of a metabolomics experiment were split and measured on both the unit-mass ISQ GC–MS and the Orbitrap GC–MS. Detected compounds were annotated (white) by EI database matching or labeled as unknowns (grey). To ensure no compounds were falsely labeled as unknowns because of low-quality spectra, only spectra with a least 20 deconvoluted fragments were considered.

**Figure 6 metabolites-10-00143-f006:**
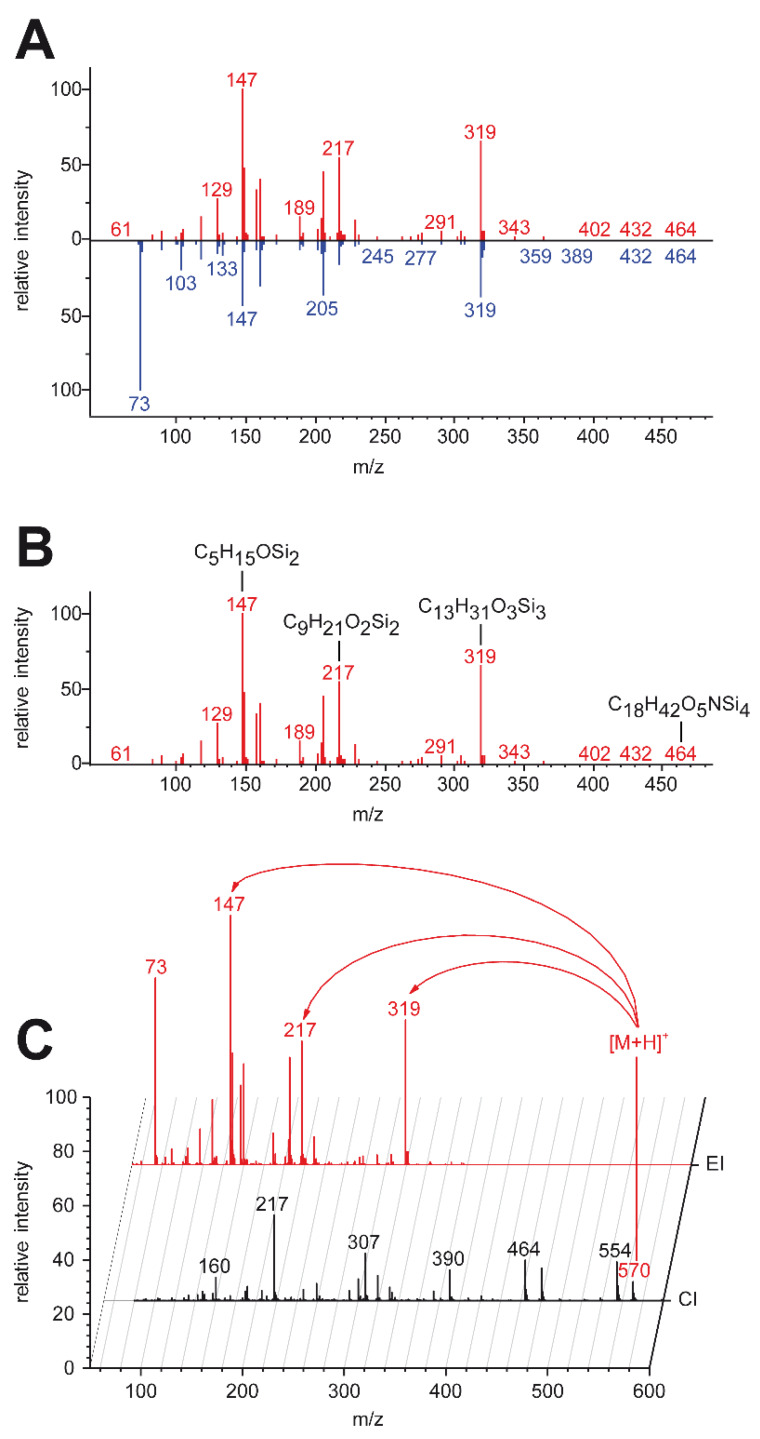
Metabolite identification workflow using high-resolution GC–MS. (**A**) EI database matching. Unit-mass EI spectra from commercially and freely available databases (blue) are compared with the unknown spectrum (red). (**B**) Calculation of sum formulas of several fragments using accurate mass capabilities. If sum formulas fit to the proposed compound by database matching, it is treated as putatively annotated, pending identification by measuring an analytical standard. (**C**) For unknown compounds with poor database matches, an accurate mass molecular ion from CI measurements (black) is assigned to the EI spectrum of the unknown (red). In silico fragmentation of candidate structures is performed and results are compared with recorded EI data [32].

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
