# Peer review of "Metabolomics Benefits from Orbitrap GC–MS—Comparison of Low- and High-Resolution GC–MS"

_metabolites, 2020, doi:10.3390/metabo10040143_

Round 1

Reviewer 1 Report

Pohnert and group present results of a toy biological experiment that was analyzed on both single-quad GC-EI-MSD and GC-EI-HRAM in an attempt to determine relative merits of the relatively new GC-Orbitrap platform.  While this study is well-intentioned and pleasingly narrow in scope, there are significant shortcomings in execution and analysis that place the manuscript on very shaky ground.  The conclusions drawn are not well substantiated by the document as currently presented. 

Mass spectrometry disciplines, particularly discovery-heavy fields such as metabolomics, are frequently driven by "new kit" that becomes available.  As labs procure new equipment, when trivial (a new triple quad from vendor "X") or when of radically different design (linear ion trap to orbitrap, for example), too few unbiased experiments are performed and published to ask how platforms affect results.  To that end, a comparison of GC-EI-MS and GC-EI/CI-Orbitrap is timely and of potential value to the community.   

As the authors allude, direct comparison of spectrometers themselves, absent variations in workup and chromatographic systemes, is critical to make meaningful and interpretable conclusions.  Unfortunately, significant variations in execution of the present experiment have failed to provide a cleanly interpretable result for this very reason.  

Methods are relatively sparse for a metabolomics-method-focused manuscript, and need to be fleshed out more completely.  Despite this omission, there are fundamental problems with experiment design and execution that significant impede interpretation of this study.  

1) Different GC oven and autosampler systems are used, making this a non-direct comparison from the start.  Although data (or figures) are lacking, the authors admit that the GC-MSD system performed relatively poorly on tests of FAMEs.  This fact immediately handicaps the GC-MSD's performance.  By how much is unclear; no graphical presentation of raw data exists anywhere in the manuscript (more on that below). 

2) The authors tangentially mention that inlet geometries are different between the systems.  How so?  Are liner sizes different?  Column connection depths?  Detail is lacking here.  Significant biases and differences in instrument performance can occur if liners are of different size and chemistry/structure (glass wool versus gooseneck, Thermo versus Restek deactivation, etc).  Are both liners split/splitless, or is a PTV/multimode inlet installed on one or both systems?  On the other end of the column, the source temperatures were different.  Why was this, and how would your results be different with consistent source temps?

3) It is unclear if new liners, columns, and other consumables were installed on these systems immediately before analysis.  Were the systems used "as is", or were sources cleaned before use?  No attention is paid to the relative performance of these systems on a clean test standard (anything from octafluoronapthalene to Supelco's FAME-37 mix)?  An older "beaten up" ISQ MSD will surely not work as well as a clean instrument, Orbi or otherwise!  

4) The authors have made a best-intended, but potentially disastrous, decision to use nearly an order of magnitude (8-fold) more biological sample during work-up and analysis for the MSD versus Orbi.  This decision was based on a poorly-justified desire to have similar TICs.  This desire is wholly inappropriate.  The Orbitrap AGC / fill-time adjustment and individual ion counts from FT-detector coil provide an entirely different output from a dynode/EM in the ISQ.  This decision is disastrous as it was accomplished by stuffing more biological material into an identically-sized reaction.  Are ISQ samples being poorly derivatized and thus thermally degrading during analysis?  Are the differences in overlap between the ISQ and Orbi data due to this altered work-up?  Even if a signal-intensity normalization were desired, changing the derivitization stoichiometry seems like one of the _worst_ possible options.  In GC, more than many fields, "more is not better" is a critical concept.  How affected was chromatographic performance by loading nearly 10-fold more biology on column?  With not a single chromatogram in the manuscript, this cannot be assessed by the reader. 

5) Claiming that the Orbitrap has a higher dynamic range than an MSD is quite a stretch.  The underlying "sample and spin" approach inherent to all FT analyzers, AGC-enabled or otherwise, causes systems to (1) have an in-scan limit on dynamic range based on cell filling (ie, 10^6 ions in one scan) and (2) become effectively blind to chromatographic events when the trap (and LIT, if used) are full.  No data are presented to show increase in in-scan or between-scan dynamic range for the Orbi over ISQ; significant previous data shows that the authors' claim that 'Orbi has higher dynamic range' is opposite from reality.

6) There was clearly much manual curation of the GC-Orbi data.  In the end, the take-away message is pitched as "GC-Orbi gave 3x the features".  Multiple arbitrary cut-offs are used throughout, yet a general view of the where features were "filtered out" is provided.  Providing an idea of where features were excluded for each arm of the analysis in detail is a critical step for a comparison where significant computational tool wrangling was required.  

General comments:

The introduction provides some nice literature review on metabolomics and annotation of (LC-MS) data, but is poorly-matched to the content of the paper.  One way to justify this mismatch is the dearth of comparable direct comparisons.  The end result is a block of text with many non sequitors, generalizations, and mentions of existing issues in the field that are LC-MS specific -- without specifying that they are not relevant to the current work.  

The authors discuss a "similar separation" when interpreting their PCA plots of treated vs untreated cells.  The authors are putting significant stock on the values of the principal component variances for both assays and (perhaps?) physical shape of the resultant ellipsoids.  This is an over-fit (pardon the statistical pun).  If anything, the PCA plots show that the discriminatory power of the ISQ and Orbi are similar -- this is not something the authors address further, which is a shame.  

While the authors have stated that EI-Orbitrap analysis leads to (significant) deviations in relative ion abundance, matching scores are still used for annotation.  Further explanation of this needs to be included, as most matching algorithms are based on a combination of peak position (m/z) and height (relative abundance for a 70eV EI source).

On line 216, The authors write "it was found that out of 75 annotations from the Orbitrap GC-MS dataset () were disproven with accurate mass information"  There is critical information missing from this sentence -- the number of compounds.

The statement ending on line 221 is unsubstantiated by data.  

The paragraph starting on line 229 is designed to praise the value of GC-HRMS for unknowns analysis, but it is unclear what is added (beyond an "additional quality control", as stated).

The first paragraph of the discussion is an ill-placed and somewhat awkward literature review of instrument comparisons.  I do not see why it was included.

The authors claim that the GC-Orbi can "resolve co-eluting isobaric fragments" is incorrect and misleading.  Co-eluting isobaric fragments, by their very nature, are unresolvable in chromatographic time or m/z space.  Separating _nearly_ isobaric compounds due to the power of HRMS (ie, higher than the ISQ) is possible, but fails to take into account the significant issues of Orbitrap in-scan dynamic range.  Moreover, line 268 says "it is difficult to estimate".  Please don't mention it, then.  

Line 358: I believe you mean "figure 5", not "figure 1"

Not even a single chromatogram is included in this work.  This is generally surprising for a mass spectrometry paper, but not something necessarily required.  Chromatographs, either XICs or TICs, would have helped convince the reader that the two systems were performing similarly (ie, peak shape and width are comparable). 

What is missing, however, is a discussion of the relative sampling rate between the ISQ and Orbitrap systems.  Modern MSDs are capable of slewing their quadrupoles at tens-of-thousands of Thompson per second, achieving double-digit Hz.  At 120k resolution, a GC Orbitrap operates at ~3Hz; the effect of this difference in peak capacity and integration robustness is not discussed.  

While the motivation behind this work -- the direct comparison of GC-MSD to GC-HRMS -- is a good one, the execution of this work has fundamental shortcomings.  These issues seriously limit the validity of the comparison being made, and as such, the value of this work to the community.  

Reviewer 2 Report

This paper was a nice comparison between two different GCMS instruments, one an inexpensive commonly used single quadrupole instrument with unit resolution to an expensive, high resolution GC-Orbitrap. The authors found an increase in the number of peaks detected, both identified and unidentified using the GC-Orbitrap.

It would be beneficial to this paper if the authors made an attempt to determine if the the unknown metabolites that differ from the two platforms are just different due to their derivatization levels. From work done in this reviewers lab (unpublished) comparing two similar instruments, it was found a number of the low abundance metabolites detected by the high resolution instrument are actually just partially derivatized metabolites compared to their higher abundance fully derivatized forms. The NIST database doesn't make this easy, it does not include a number of partial derivatives. For example, only the 3 TMS version of threonine is available in the NIST database but the 2 TMS version is detected using the more sensitive instrument. Overall, an interesting comparison.

Line 92. Methoxylating, not methoxymating

Line 142, two in in

Author Response

This paper was a nice comparison between two different GCMS instruments, one an inexpensive commonly used single quadrupole instrument with unit resolution to an expensive, high resolution GC-Orbitrap. The authors found an increase in the number of peaks detected, both identified and unidentified using the GC-Orbitrap.

It would be beneficial to this paper if the authors made an attempt to determine if the the unknown metabolites that differ from the two platforms are just different due to their derivatization levels. From work done in this reviewers lab (unpublished) comparing two similar instruments, it was found a number of the low abundance metabolites detected by the high resolution instrument are actually just partially derivatized metabolites compared to their higher abundance fully derivatized forms. The NIST database doesn't make this easy, it does not include a number of partial derivatives. For example, only the 3 TMS version of threonine is available in the NIST database but the 2 TMS version is detected using the more sensitive instrument. Overall, an interesting comparison.

We have now added a comment about different derivatization products in the manuscript (Line 417ff). When we did see multiple peaks in the analysis of a pure standard, we compared all of them to the unknowns and have identified some as additional derivatization products. For example, proline standard gave us 3 peaks – the normal 2TMS version, an unexplained (but annotated by NIST) proline+CO2 2TMS and a completely unannotated peak that we also found as an unknown.

We believe that such analyses would go beyond the scope of the manuscript, which is not focusing on individual structures.

Line 92. Methoxylating, not methoxymating

Corrected.

Reviewer 3 Report

This manuscript by Stettin, et al., describes the various advantages of GC-Orbitrap MS technologies in comparison with single quadrupole GC-MS, using osmotic stress treatment of a microalga as a specific example. The high-resolution instrument picks up more compounds (157 vs. 56) and more of these—absolute number, not percentage—can be annotated (75 vs. 43). There is one specific result, namely that dehydroascorbate could be identified with the high-resolution instrument and not with the single quad GC-MS.

The results are perhaps not groundbreaking, yet the present work is entirely appropriate for publication in Metabolomics, especially having perused various accepted articles in this journal. In fact, metabolomics contains many metabolomics studies of this sort, including many studies focused on emerging technologies. Thus, I recommend acceptance of the study in its present form.

The manuscript is highly detailed and exquisitely written. There are no errors that I can spot in terms of grammar, spelling or syntax.

Author Response

This manuscript by Stettin, et al., describes the various advantages of GC-Orbitrap MS technologies in comparison with single quadrupole GC-MS, using osmotic stress treatment of a microalga as a specific example. The high-resolution instrument picks up more compounds (157 vs. 56) and more of these—absolute number, not percentage—can be annotated (75 vs. 43). There is one specific result, namely that dehydroascorbate could be identified with the high-resolution instrument and not with the single quad GC-MS.

The results are perhaps not groundbreaking, yet the present work is entirely appropriate for publication in Metabolomics, especially having perused various accepted articles in this journal. In fact, metabolomics contains many metabolomics studies of this sort, including many studies focused on emerging technologies. Thus, I recommend acceptance of the study in its present form.

The manuscript is highly detailed and exquisitely written. There are no errors that I can spot in terms of grammar, spelling or syntax.

 We thank the reviewer for the comment.

Line 142, two in in

Corrected.

Reviewer 4 Report

Overall this was an interesting read and it's always good to see a solid comparison of two different instrumentation setups. That said, I think a change of focus for the paper might be useful. i) - it might be worth being a little more specific about the applications - while loading 8 times as much would be fine for biotech or environmental applications, it's more difficult for clinical, and therefore maybe not a fair comparison (although the GC-Orbi still wins!). ii) The paper seems a little unsure whether it's trying to be a straight comparison of the two instruments' capabilities (especially with statements narrowing some quantitative reproducibility parameters down to the GC instrument and transfer tubing) or not, so perhaps the focus (and title) should be changed to a typical low res workflow compared to a high res workflow - this would make the caveats about the comparison a little less major and allow the authors more scope to explore the suitability of the two systems in the discussion, especially bringing the relative costs, sample sizes, data analysis pipelines, etc of the two workflows into consideration.

I also had some specific comments that I've covered line by line.

Lines 51-61 – the reference to the libraries is, of course, true, but the GC-Orbitrap’s spectral output, while sharing the mass profile of the libraries, doesn’t have the same intensities, making the matches weaker. Of course, as higher resolution instruments become more commonplace and the databases expand this becomes less of an issue, but it isn’t quite the same benefit that the authors describe. This is described a bit better in the discussion but perhaps worth saying up front.

Line 76 – This is an odd statement – do the authors have a reference or data to back this up? Are the quadrupoles more leaky on the ESI orbitraps than the GC instrument? Surely this depends on the isolation window and hence the RF and DC bias on the quad?

Line 77-81 I’m not sure this is a drawback – indeed it is a particular benefit of EI-based instruments that they produce richer fragmentation patterns than the average ESI-CID experiment.

Line 82-83 – Yes this is a particularly nice feature of the instrument’s software.

Line 131 – This is an interesting technique, but I would take a bit of convincing for it to make much sense – the detection method is completely different on both instruments, so to fix both to the same arbitrary number of counts seems odd. Would S/N of a particular ion not have been better? That said – I take the author’s point that this is supposed to be an examination of the ‘equivalent’ experiment, not a direct comparison. The experiment is not sample limited, which would be a concern for, for example serum or CSF analysis, but not for what I take to be an environmental analysis. Can the authors comment on the typical sample amounts used? Were they experimentally equivalent? Perhaps better to focus on ‘a typical metabolomics workflow and sample types’ on both instruments.

Line 178 – This is not really quantitative accuracy, it’s reproducibility of QC samples. If it were quantitative accuracy I would expect to see a panel of standards, providing their linearity over full dynamic range, as well as the error over the same range. If this information is available, that’s great, but otherwise best to change the section heading.

Lines 190-199 – this is actually quite a nice bit of information.

Lines 207-218 – Can the authors please comment on the 339 ‘compounds’ detected in 2.1 and the 75 annotations from the library – what happened to the other 254 compounds? There’s an allusion in the discussion to them not being identified but that potentially makes the more interesting. Can the authors comment on why the library match rate was so poor?

References – the authors should probably cite our paper (Metabolomics volume 12, Article number: 189 (2016)) which I think was the second GC-Orbi metabolomics paper after Josh Coon’s, and the first comparative one.

Round 2

Reviewer 1 Report

I was surprised to see this manuscript back in a month's time, particularly with so few changes.  My original recommendation (and the sole reviewing recommendation, it would appear) was "reject".  

I spent a good amount of time writing both a summary and point-by-point review of the original submission; perhaps this was in error, as my intent was to explain some of the fundamental flaws in the submission, not provide a roadmap and check-list for resubmission.

I appreciate the authors' tenacity and minor edits to the text.  The inclusion of details about the single quadrupole instrument and its configuration further solidified some of my concerns that this is an irrelevant comparison.  As I said in my first review, this is a fundamentally flawed study.  This work is not of generalizable use to the community, does not reflect a large body of work that warrants embodiment in a "journal of record" and, most worryingly, makes conclusions that could be generalized based on an improper design.

An analogy follows:

Title: "Digital photo editing benefits of the iPad Pro-- an in-depth comparison to Windows PCs". 

Abstract (shortened): iPad Pro is a new system that has not been used much for photo editing.  Windows PCs are more commonly used.  We compared the performance of iPad Pro and Windows PCs by editing the same two files in Photoshop on each platform. We found that iPad Pro is 3x faster on average for all tasks, and that the iPad Pro's touch-screen let us do things we couldn't do on the Windows PC.  Overall, the iPad Pro is a great tool that outperforms Windows PCs, although there is a need for better ways to handle file output.  

Sounds great, until one digs in a bit deeper and finds that the authors made a comparison was made between a modern iPad Pro and a 10-year-old antiquated Windows PC.  I'm sure Apple would love to cite such a story out of context, but doing so would be dangerously disingenuous and not reflect a fair or accurate comparison of current state of the art. 

That's what you've done here.  This is not a systematic comparison, it's a single case study of what's in your lab.  This is not a "single quad versus orbitrap" in an apples-to-apples comparison.  This is, more to the point, not an "in-depth comparison to GC singlequad", it's a one-off comparison to the single quad that's in your lab. 

This is not a story that warrants publication.   

Author Response

We thank the editor for asking an additional reviewer after receiving two biased reports from reviewer 1. We are very happy to see that the fourth reviewer strongly supports publication as reviewer 2 and 3 did.

In our revisions and reply, we focus on the additional reviewer as recommended by the editor.

Reviewer 4 Report

With the refocusing of the paper to compare a typical workflow on a single quadrupole instrument versus a GC-orbitrap, the work presents a far clearer story than previously.

My other comments have been dealt with effectively, with clarification or removal of ambiguous or confusing statements.

The new discussion of validated unknowns is interesting. I would perhaps suggest a slight expansion of the newly added sentence in line 256/257, where the authors state that there is no additional benefit from HR-MS in verification. Since accurate masses and therefore predicted formulae for the fragment ions are available, could this shed any light on the nature of these compounds? One would predict in natural product workflows where dereplication is required, that this method would be extremely exciting!

Otherwise, I see no reason why the article can't be published in its current form.

Author Response

With the refocusing of the paper to compare a typical workflow on a single quadrupole instrument versus a GC-orbitrap, the work presents a far clearer story than previously.

My other comments have been dealt with effectively, with clarification or removal of ambiguous or confusing statements.

The new discussion of validated unknowns is interesting. I would perhaps suggest a slight expansion of the newly added sentence in line 256/257, where the authors state that there is no additional benefit from HR-MS in verification. Since accurate masses and therefore predicted formulae for the fragment ions are available, could this shed any light on the nature of these compounds? One would predict in natural product workflows where dereplication is required, that this method would be extremely exciting!

Line 256 is dealing with the overlap of dysregulated compounds on 2 instruments. We assume the comment refers to line 285f:

We agree that predicted sum formulas could shed light on the nature of unknown compounds, if computational tools were available supporting it, like high-resolution molecular networking that exists for LC-MS. Right now, just sum formulas of fragment ions even combined with an MS expert looking at them seem to be not enough to gain full structural information about the unknown. The only alternative right now is attempting a complete structural elucidation with advanced techniques involving molecular ions from CI, compound libraries and in silico fragmentation tools, like it has been done in this study. We agree that there are needs for a solution bridging standard libraries and full structural elucidation especially for the purpose of dereplication. We have expanded the paragraph in the manuscript to include this notion.

Otherwise, I see no reason why the article can't be published in its current form.

We thank this reviewer for the supportive and constructive statements